# In Vitro Mucoadhesive Features of Gliadin Nanoparticles Containing Thiamine Hydrochloride

**DOI:** 10.3390/pharmaceutics16101296

**Published:** 2024-10-04

**Authors:** Silvia Voci, Agnese Gagliardi, Elena Giuliano, Maria Cristina Salvatici, Antonio Procopio, Donato Cosco

**Affiliations:** 1Department of Health Sciences, University “Magna Græcia” of Catanzaro, Campus Universitario “S Venuta”, 8100 Catanzaro, Italy; silvia.voci@unicz.it (S.V.); gagliardi@unicz.it (A.G.); elena.giuliano@unicz.it (E.G.); procopio@unicz.it (A.P.); 2“AGreenFood” Research Center, University “Magna Græcia” of Catanzaro, Campus Universitario “S Venuta”, 88100 Catanzaro, Italy; 3Institute of Chemistry of Organometallic Compounds (ICCOM)-Electron Microscopy Centre (Ce.M.E.), National Research Council (CNR), via Madonna del Piano n. 10, Sesto Fiorentino, 50019 Firenze, Italy; salvatici@ceme.fi.cnr.it

**Keywords:** gliadin, mucin, nanoparticles, oral delivery, thiamine hydrochloride

## Abstract

Background: Gliadins have aroused significant interest in the last decade as suitable biomaterials for food and pharmaceutical applications. In particular, the oral route is the preferred method of administration for gliadin-based formulations, due to the affinity of this biomaterial for the gut mucosa. However, up to now, this has been demonstrated only by means of in vivo or ex vivo studies. Methods: This is why, in this study, various in vitro techniques were employed in order to evaluate the ability of polymeric nanoparticles, made up of a commercial grade of the protein and an etheric surfactant, to interact with porcine gastric mucin. The nanosystems were also used for the encapsulation of thiamine hydrochloride, used as a model of a micronutrient. Results: The resulting systems were characterized by a mean diameter of ~160–170 nm, a narrow size distribution when 0.2–0.6 mg/mL of thiamine was used, and an encapsulation efficiency between 30 and 45% of the drug initially employed. The incubation of the gliadin nanosystems with various concentrations of porcine gastric mucin evidenced the ability of the carriers to interact with the mucus glycoprotein, showing a decreased Zeta potential after a 4 h incubation (from ~−30 to −40 mV), while demonstrating that the encapsulation of the drug did not affect its bioadhesive features. Conclusions: Altogether, these data support the conceivable application of gliadin nanoparticles as formulations for the oral administration of bioactive compounds.

## 1. Introduction

The adhesive phenomena occurring between a biomaterial and a biological substrate are generally referred to as bioadhesion [1]. When the interaction involves the mucus layer covering the epithelium of the various organs and cavities of the human body, mucoadhesion is the most appropriate term to use [2].

Mucus represents a complex fluid made up of a large amount of water, but also lipids, ions and mucins, the glycosylated proteins responsible for its viscoelastic behavior [3]. These macromolecules are characterized by repetitive proline–threonine–serine domains (PTS), which act as anchors for the O-glycosylation reaction involving the attachment of several sugar moieties (from 5 to 15 residues) with sialic acid, fucose or galactose as the terminal units [4,5]. These polar terminal chains are known to assemble according to a bottle-brush conformation, and confer to mucin a net negative surface charge, due to the presence of free sulfonated and carboxylate groups [6].

At the same time, the structure of mucin is characterized by lightly or non-glycosylated portions rich in cysteine residues (cysteine knots), responsible for the assembly of the glycoprotein fibers into a tridimensional porous network through non-covalent interactions [4,7].

Notwithstanding the inherent differences in terms of the pH, thickness and mesh size that characterize the mucus of a body compartment, its primary role is to act as a protective barrier that helps to trap any potential harmful entity entering the host body by means of size exclusion and flow mechanisms, respectively [8,9]. These intrinsic features of the mucus matrix have been exploited, for example, in oral drug delivery, in order to customize the interaction rate of the investigated formulation with the mucus components by means of mucoadhesive or mucopenetrating carriers, with the aim of achieving sustained drug concentrations or improving the uptake of the cargo molecule by the absorptive epithelium [10,11,12,13].

In this context, the interactions between mucin and colloidal formulations can be investigated by means of in vitro, in vivo or ex vivo assays; the former ones are cost and time saving with respect to other techniques, and they avoid the ethical constraints related to the use of animal models or biological tissues [14,15,16]. Among these, dynamic and electrophoretic light scattering (DLS and ELS, respectively), as well as turbidimetry analyses, have been widely exploited in order to evaluate the bioadhesive features of biomaterials, as was true, for example, for polysuccinimide- [16], chitosan- [17] and zein-based nanoparticles [18].

Herein, these techniques were used in order to investigate the in vitro mucoadhesive properties of gliadin nanoparticles (GNPs). This prolamin, obtained from wheat grains, has been used for the development of various nanoformulations in the pharmaceutical [19,20,21] and alimentary fields [22,23,24].

Although previous works have demonstrated the in vivo bioadhesive potential of this biomaterial [25,26,27,28,29,30], to the best of the authors’ knowledge, this is the first investigation aiming at characterizing mucin–gliadin interactions by using an in vitro approach.

The nutritional and physiological relevance of thiamine (B1) is well known, being involved in the oxidative and non-oxidative metabolism of carbohydrates as well as in the homeostasis of neuronal cells [31]. However, heat, exposure to light and the oxygen sensitivity of the molecule all significantly affect its long-term stability and compromise its use, as is true for other vitamins [32]. At the same time, its scant bioavailability (~5%) is the main issue associated with the oral administration of B1 [33]. In this regard, several carriers that are able to retain B1 have been employed to address the chemical and physical instability of this molecule, obtaining novel formulations for pharmaceutical [34,35] and food applications [36,37,38,39,40,41,42]. However, few investigations have described the possibility of using proteins as raw materials for the encapsulation of B-complex vitamins; for example, vitamin B12 was entrapped within soy protein isolate [43], whereas only recently have zein- [44] and gliadin- [45] based core–shell microcapsules been used to retain B1.

Given this state of the research, in this work, nanoparticles made up of gliadin and stabilized by Brij O2, previously developed by our research team [46], were incubated with various concentrations of mucin and their physico-chemical properties were investigated by DLS, ELS and turbidimetry. Finally, thiamine hydrochloride (B1) was encapsulated within GNPs, and the influence of the micronutrient on the interaction between the nanosystems and mucin was evaluated.

## 2. Materials and Methods

### 2.1. Materials

Gliadin from wheat (CAS number 9007-90-3), mucin from porcine stomach (CAS number 84082-64-4, type II, bound sialic acid ≤1.2%), thiamine hydrochloride (B1, CAS number 67-03-8, reagent grade ≥99%, molecular weight 337.27 g/mol), pepsin from porcine gastric mucosa (≥400 units/mg of protein) and pancreatin from the porcine pancreas (≥3 × USP specifications) were purchased from Sigma-Aldrich (Merk Life Science S.r.l., Milan, Italy). Polyoxyethylene (2) oleyl ether (Brij O2, CAS number 9004-98-2, molecular weight 356.58 g/mol) was obtained from Croda International (Snaith, UK), while ethanol was provided by Carlo Erba S.p.A (Rodano, Milan, Italy).

### 2.2. Preparation and Physico-Chemical Characterization of Brij GNPs

Gliadin-based carriers were obtained as previously described [46,47]. Briefly, the protein (0.17% *w*/*v*) and Brij O2 (0.1% *w*/*v*) were dissolved in 3 mL of an organic phase made up of 3 mL of a hydroalcoholic solution (EtOH:H_2_O 7:3 *v*/*v*, pH 10) and then added to an aqueous phase consisting of 5 mL of MilliQ water. The obtained mixture was homogenized for 2 min and stirred on a magnetic plate for 6 h at room temperature in order to favor ethanol evaporation (Figure 1) [46,47]. The final gliadin concentration was 0.1% *w*/*v*.

B1-loaded GNPs were obtained using the same approach by solubilizing different concentrations of the compound in the aqueous phase of the formulation (0.2–0.8 mg/mL). Successively, the nanosystems were purified by ultracentrifugation (Optima MAX-XP apparatus, Beckman Coulter Life Sciences, Brea, CA, USA) at 90k g (1 h, 4 °C) [48].

The mean sizes, polydispersity index and surface charge of the obtained nanosystems were evaluated by DLS using a Zetasizer Nano ZS (Malvern Panalytical Ltd., Spectris plc, UK) by diluting each sample in bidistilled water at a 1:50 *v*/*v* ratio [49]. This technique was also employed in order to evaluate the physico-chemical stability of the samples under simulated gastrointestinal conditions [50], while the morphology of the GNPs was investigated by transmission electron microscopy (TEM), as previously reported [51].

The in vitro mucoadhesive features of the samples were evaluated following their time-course incubations (2 and 4 h) in different aqueous solutions of mucin (0.1, 0.25 and 0.5% *w*/*v*), as previously reported [52,53].

Moreover, the stability of the GNPs was investigated by using a Turbiscan Lab Expert^®^ apparatus (Formulaction, Toulouse, France), as previously reported [48,54]. In detail, the variations in the backscattering and transmission profiles (ΔBS and ΔBT, respectively) of the nanoparticles were analyzed as a function of time and incubation temperature; then, the results were expressed using the Turbiscan Stability Index (TSI) and processed with Turbisoft 2.0 software [48,54].

### 2.3. Fourier-Transformed Infrared (FT-IR) Analyses

The IR spectra of gliadin, mucin and thiamine hydrochloride (as powders), as well as those of Brij O2 (in the liquid state), were collected and analyzed as single raw materials and as physical mixtures and, once assembled into NPs (as freeze-dried powders), by means of a Nicolet iS5 Spectrometer equipped with a diamond attenuated total reflectance device (Thermo Fisher Scientific, Waltham, MA, USA). The IR profiles were recorded in the transmittance mode between the 4000 and 600 cm^−1^ region with a resolution of 4 cm^−1^, and are the average of 64 scans; data interpretation was carried out by using Omnic 9.12.1019 software (Thermo Fisher Scientific, Waltham, MA, USA), as previously described [55].

### 2.4. Entrapment Efficiency, Loading Capacity and Release Profiles

The amount of B1 retained by the carriers was quantified by UV-vis spectroscopy (Lambda 35, Perkin Elmer, Waltham, MA, USA) at the λmax of 246 nm [45].

In detail, each formulation was centrifuged as previously described (see Section 2.2), and the supernatant obtained at the end of the process was analyzed at the aforesaid λmax; the entrapment efficiency (EE%) was calculated by plotting the amount of compound that became entrapped within the polymeric matrix (De) against the amount of thiamine hydrochloride initially added during the preparation procedure of the NPs (Dt), as follows:EE (%) = De/Dt × 100(1)

An empty formulation was used as a blank.

The amount of stabilizer integrated into the GNPs was calculated by using the iodine–iodide assay as previously described [56]; then, the drug-loading capacity (LC%) was expressed as the ratio between the amount of the entrapped B1 versus the total weight of the formulations, as follows:LC (%) = (Entrapped drug)/(Total weight of NPs) × 100(2)

Moreover, the release kinetics of the compounds were investigated under simulated gastric (SGF, pH 1.2) and intestinal fluids (SIF, pH 6.8), respectively, as previously reported with slight modifications [57]. In detail, 1 mL of each sample was placed in a cellulose dialysis bag (cutoff 10 kDa, Spectrum Laboratories Inc., Eindhoven, Netherlands), sealed at both ends with clips and immersed in 100 mL of release fluids [57].

At fixed incubation times, 1 mL of the receptor fluid was collected and replaced with the same amount of fresh medium in order to maintain the sink conditions; the various aliquots were analyzed by UV-vis spectroscopy at the λ_max_ previously reported, and according to the following equation:Release (%) = drug_rel_/drug_load_ × 100(3)
where drug_rel_ is the amount of the drug released at time t and drug_load_ is the amount of the drug entrapped within the GNPs.

### 2.5. Adsorption of Mucin

The amount of mucin adsorbed onto the surface of nanosystems upon incubation (see Section 2.2) was calculated by UV-vis spectroscopy according to the method of Boya and coworkers [58]. In detail, a standard calibration curve for mucin was obtained by diluting a 1 mg/mL stock solution of the protein in water (600–1.35 μg/mL); the obtained calibration curve was
y = 0.004791x + 0.003091(4)
where x represents the known mucin concentration (µg/mL) and y the relative absorbance, with a coefficient of linear regression (r^2^) equal to 0.9998. For the analysis, each sample was centrifuged as described in Section 2.2, and the supernatant was analyzed at a wavelength of 280 nm; the amount of mucin bound to the particle surface was calculated as follows:Mucin bound (%) = Mb/Mt × 100(5)
where Mb and Mt represent the amount of glycoprotein that was adsorbed onto the NPs and the amount of mucin initially added, respectively. No interference between the mucin and the other components of the nanoformulations was observed.

Moreover, the mucoadhesive properties of the samples were also investigated by turbidimetric measurements [12,59]. In detail, 0.5 mg/mL of the samples were incubated with various mucin solutions (0.1, 0.25 and 0.5% *w*/*v*) and their absorbance (Abs) was measured at 500 nm, and the results were expressed as the % of turbidity increase according to the following equation:Turbidity increase % = (AbsGNPs + mucin − Abscontrol) × 100/AbsNPs(6)

The Abs of pure mucin was used as the control. An increase in the relative Abs values demonstrated the interaction between the GNPs and the glycoprotein [12,59].

### 2.6. Statistical Analysis

The statistical analyses of the various experiments were performed by one-way ANOVAs and the results were confirmed by a Bonferroni *t*-test, with a *p* value of <0.05 considered statistically significant.

## 3. Results and Discussion

### 3.1. Characterization of Brij O2-Stabilized GNPs Containing B1

Previous investigations carried out by our research team have demonstrated the contribution of Brij O2 to the stabilization of GNPs, characterized by physico-chemical properties suitable for use as delivery systems for different bioactives [46,47], as well as suitable safety profiles against normal and tumor cells for up to 25 μg/mL of the biopolymer [46,47]. In addition, it has been shown that the in vivo administration of these nanosystems promotes the targeting of the bowel compartment, probably as a consequence of their mucoadhesive features [30].

Starting from these findings, the goal of this investigation was to provide a comprehensive evaluation of the interactions occurring between GNPs and a commercial grade of mucin by using various techniques. In addition, the influence of the encapsulation of B1 within GNPs, used as a model compound, on the aforesaid features was also investigated.

In particular, the first tests of this work aimed at selecting the best drug-loaded formulation for use in the following mucoadhesion experiments (Figure 1A–C). In detail, the GNPs containing B1 evidenced mean sizes between 160 and 170 nm, slightly increased with respect to the empty formulation (~140 nm), and concentrations > 0.6 mg/mL of B1 evidenced the formation of macroaggregates and a polydispersed population (PdI > 0.3). The surface charge of the GNPs decreased from −32 to −45 mV when 0.4 mg/mL of B1 was added (Figure 1A–C), while their morphology was not influenced by the addition of the drug, confirming their spherical shape and smooth surface (Figure 1D,E).

Successively, the physical time- and temperature-dependent stability of the samples was investigated by means of the multiple light scattering technique. The variation in the TSI profiles of the nanosystems containing B1 was monitored and compared with that of the empty formulation (Figure 2). As shown in Figure 2, the samples prepared with increasing concentrations of the bioactive evidence an overlapping TSI profile when the analyses were performed at room temperature; this trend was also confirmed by the ΔBS and ΔT profiles reported in Appendix A. However, when the test was performed at 37 °C, a sharp increase in the TSI slope of the formulation prepared with an initial drug concentration of 0.8 mg/mL is observed (Figure 2). The obtained results demonstrate that the increase in temperature promoted a certain degree of destabilization of this sample, probably due to flocculation or the creaming phenomena, as previously described [60,61,62] (Appendix A). The TSI profiles of the various samples stored at 4 °C for 28 days confirmed this trend (Appendix A), evidencing negligible variations in the kinetic profiles of the empty systems and the formulations prepared with a drug concentration of up to 0.6 mg/mL, while an increase in this parameter was shown after two weeks for the nanoparticles prepared with an initial amount of B1 of 0.8 mg/mL (Appendix A).

The encapsulation profiles of B1 within the Brij O2 GNPs evidenced a progressive decrease in the retention rate of the drug (Figure 3A); in more detail, the highest EE% was obtained when 0.2 mg/mL of B1 was added during the preparation procedure (~45% of the drug used became entrapped, 0.089 mg/mL). On the contrary, this value decreased to 25 and ~30% when the formulations were prepared with an initial drug concentration equal to 0.4 and 0.8 mg/mL, respectively, corresponding to 0.1 and 0.24 mg/mL of B1 integrated within the carriers, respectively (Figure 3A). The evaluation of the LC% evidenced values of 13% when 0.6 mg/mL of B1 was initially used for the preparation of the nanosystems, and similar values (15%) were obtained when 0.8 mg/mL of thiamine was used; this probably occurred because the maximum amount of the compound that could be held by the carriers was reached. The data are in agreement with that previously discussed (Figure 1A).

The evaluation of the release profiles of B1 from the GNPs confirmed the ability of the protein matrix to provide the controlled leakage of the entrapped bioactive [22,45,57], as well as their peculiar trend of releasing their payload in a manner that was inversely proportional to the initial drug concentration used during the preparation of the samples (Figure 3B) [46,47]. In detail, an initial release of 30–50% of B1 was obtained under SGF that was followed by a slower leakage of the compound when the pH was 6.8 (Figure 3B). This second phase of release can be advantageous, considering the possibility of improving the diffusion of the drug-loaded nanosystems towards the intestinal epithelium once administered in vivo (Figure 3B).

In view of these results, the formulation prepared with an initial B1 concentration of 0.6 mg/mL was chosen as the most promising nanosystem to be potentially used for the oral supplementation of B1. This is why additional tests aimed at investigating the stability of the proposed GNPs under simulated gastrointestinal conditions were performed. For this reason, the empty nanosystems and GNPs containing B1 were incubated in simulated gastric (SGF) and intestinal fluids (SIF) enriched with pepsin and pancreatin, respectively, and their physico-chemical properties were evaluated.

As reported in Appendix A, it was found that after a 2 h incubation in acidic conditions, a certain amount of aggregation of the nanoparticles occurred, with PdI values ≈ 0.4 and an increase of two times their mean sizes was noted. The use of pancreatin and a basic pH promoted an additional increase in the polydispersity index and a decrease in the particle diameter due to the proteolytic activity of the SIF, as reported for other protein-based nanosystems [50,63,64,65] (Appendix A). This activity was also confirmed by the values of the Zeta-potential that were close to neutrality at the end of the incubation time (Appendix A).

Finally, the FT-IR technique was used in order to provide qualitative information concerning the interactions occurring between the various components of the formulation (Table 1 and Figure 3C) [55,66]. The spectrum of B1 evidenced the characteristic adsorption bands of the pyrimidine ring between 3500 and 3400 cm^−1^, as well as between 1603 and 1384 cm^−1^, which represent the stretching of the free amine group and the vibrations of the N-H bending and C-N inside the heterocycle, respectively [67] (Figure 3C). In addition, at 2940 and 1359 cm^−1^ the peaks of the -CH_3_ groups occurring in the pyrimidine and thiazole rings were found, respectively, while at ~3250 and 1060 cm^−1^ the stretching of the -OH and C-O groups of the ethanol moiety linked to the thiazole heterocycle of B1 were detected. Finally, the stretching of the C-S-C group was located at 770 cm^−1^ [67,68] (Table 1 and Figure 3C).

After the encapsulation procedure, it was found that the signals observed in the 3000–3400 cm^−1^ region and related to the pyrimidine ring of B1 were completely masked by the Amide I of the protein; in addition, the stretching of the C-18 chain of the Brij previously occurring with two bands at 2920 and 2843 cm^−1^ and with two other peaks in the 700–800 cm^−1^ region were now merged (Table 1 and Figure 3C). These changes were followed by the consistent displacement of the Amide III signal of the empty nanosystems (from 1450 to 1435 cm^−1^), along with the disappearance of the bands at 1352 cm^−1^ and those related to the -NH bending and C-O stretching of the thiazole ring of B1 previously described. Altogether, this evidence allows us to conclude that the entrapment of thiamine within GNPs is driven by polar and hydrophobic interactions (Table 1 and Figure 3C).

Altogether, the information reported in this section was employed for the selection of the formulation to be used for the in vitro mucoadhesion studies that will be described further on. Specifically, this was the gliadin sample obtained using an initial thiamine concentration equal to 0.6 mg/mL, which evidenced suitable stability as well as ideal physico-chemical and technological features with respect to the various conditions tested. 

### 3.2. In Vitro Mucoadhesive Features of B1-Loaded Brij O2 GNPs

The evaluation of the mean sizes and Z-potential of the NPs incubated with mucin represents an easy and well-established way to characterize their mucoadhesive properties [6,69,70].

The DLS analyses were performed by incubating the GNPs (0.5 mg/mL) with increasing concentrations of mucin (0.1, 0.25 and 0.5% *w*/*v*) in order to evaluate their interaction. The data reported in Figure 4 show that the empty GNPs evidenced an increase in their mean diameter almost proportional to the concentration of mucin in the incubation medium, and this trend was maintained after various incubation times (Figure 4). The evaluation of the surface charge of 0.5 mg/mL of the GNPs evidenced a decrease from −28 mV to ~−36/−40 mV after a 4 h incubation, suggesting the ability of mucin to be adsorbed onto the surface of colloidal systems (Figure 4).

The fact that the tested formulations were characterized by a negative surface charge in their native state did not preclude their bond with the mucin molecules, as has already been shown for iron oxide NPs decorated with Pluronic F127 and β-cyclodextrin [58] itraconazole-loaded NPs made up of different PEG derivatives [6] and bovine serum albumin carriers [71] (Figure 4).

The formulations containing B1 showed a similar trend, although the same decrease in the Z-potential was observed just after a 2 h incubation, and the obtained values remained constant over the time (Figure 4); moreover, an increase in the sizes of the GNPs containing B1 with respect to the empty systems was evident after a 4 h incubation, probably because of the adsorption of a great number of mucin molecules (Figure 4).

In order to corroborate these data, the amount of mucin adsorbed by the nanoformulations upon their incubation within a medium prepared with an initial amount of glycoprotein equal to 0.5% *w*/*v* was quantified by spectrophotometric analyses (Figure 5A); these showed that the presence of the drug slightly influenced the ability of the nanocarriers to retain the glycoprotein; in fact, 3000–3200 μg/mL of mucin were bound onto the surface of the empty GNPs, while this value ranged between 3000 and 3400 μg/mL for the nanosystems containing B1 (Figure 5A). Furthermore, after a 4 h incubation, the amount of mucin integrated onto the empty and B1-loaded carriers remained constant, probably as a consequence of the saturation of their surface [10] (Figure 5A).

Figure 5B and Table 2 show the FT-IR spectrum of the freeze-dried formulations after a 4 h incubation with 0.5% *w*/*v* of mucin. Herein, the glycoprotein evidenced multiple low-intensity bands in the 3800–3560 cm^−1^ region due to the vibrations of the amine, amide and alcoholic groups [72]; these signals were followed by peaks at 2920 cm^−1^ (symmetric stretching of -CH_2_ groups) [73] and those related to the Amides I (1635 cm^−1^) and II (1540 cm^−1^), as well as to a band at 1225 cm^−1^ resulting from the N-H in plane bending [73,74]. Finally, the intense peak observed at 1033 cm^−1^ represents the pyranose ring of the N-acetyl-glucosamine residue (Figure 5B) [74].

A broad and intense band at 3281 and 3293 cm^−1^ for the empty and B1-GNPs, respectively, confirmed the adsorption of mucin onto the colloidal surface of the nanosystems due to several -NH and -OH residues occurring within the structure of each protein; this was also followed by a consistent reduction in the intensity of the peak related to the amino sugar moieties of mucin (Figure 5B). The interaction between the glycoprotein and the nanoformulations was also supported by the disappearance of the peaks related to the Amide II signal of the Brij O2 GNPs and the amino sugar moiety of mucin, together with the blueshift of the Amide III frequencies of the gliadin samples from 1347 to 1353 cm^−1^ for the empty nanosystems, and from 1345 to 1302 cm^−1^ for the B1-loaded GNPs (Figure 5B).

The numerous lipophilic and neutral aminoacidic residues that gliadin contains are responsible for its mucoadhesive properties [75,76]; moreover, they promote non-covalent and polar interactions with mucin throughout the gastrointestinal tract. This aspect was investigated by turbidimetric assays performed on the mucin–NPs complexes upon 4 h of incubation using pure mucin as the control (Appendix A); this was done because turbidity measurements can quickly assess the interaction between mucoadhesive biomaterials and a protein [77]. As can be seen, when 0.5 mg/mL of the samples were added to the mucin solution, the highest increase in terms of turbidity was observed with respect to the control (pure mucin) when they were incubated with 0.5% *w*/*v* of the glycoprotein, suggesting that, as previously observed (Figure 4) and as demonstrated by [78], this ratio was the one where the interactions were maximized (Appendix A).

## 4. Conclusions

The interaction rate between mucin and the nanoparticles proposed for oral administration provides useful information about the potential in vivo fate of the colloidal systems. In addition, the opportunity to exploit nanotechnology to deliver essential micronutrients is a useful strategy aiming at increasing their bioavailability and stability, and improving the sensory features of compounds characterized by an unpleasant flavor or smell, as is true for vitamins belonging to the B complex [79].

In this work, DLS, ELS and turbidimetry were used in order to characterize gliadin nanoparticles stabilized with Brij O2 (0.1% *w*/*v*), examining their interaction with mucin. The systems demonstrated the retention of thiamine hydrochloride, showing that hydrogen bonding is the main phenomenon governing the entrapment of the drug. The incubation of the GNPs with mucin confirmed the mucoadhesive properties of the polymer as demonstrated by the decrease in the Z-potential values, while the encapsulation of B1 promoted a slight increase in the amount of mucin that became bound onto the surface of the carriers.

In conclusion, the findings provided in this manuscript represent the starting point for the development of a nanoformulation that could be used for the oral supplementation of B1; in this regard, the fact that GNPs are made up of low-cost raw materials and that they can be obtained by a rapid and simple preparation procedure may be useful for the scale-up of the proposed nanosystems. Moreover, one of the main advantages derived from the nanoencapsulation of B1, with respect to the existing conventional dosage forms available on the market, is the possibility of achieving a sustained and controlled release over the time.

B1 is orally absorbed by means of saturated active transport mechanisms mediated by THTR-1 and -2 transporters and passive diffusion in the small intestine [33,80]; considering this aspect and the fact that, as recently demonstrated [30], at 4 h post-oral administration, Brij O2 GNPs are able to reach the small intestine and specifically accumulate in the ileum compartment, the proposed formulation is promising for an in vivo oral delivery system for B1. Nonetheless, investigations aimed at evaluating the relative oral bioavailability and pharmacokinetic profiles of B1-loaded GNPs, with respect to the existing conventional dosage forms, need to be performed in order to propose the described systems as plausible B1 supplementation formulas.

## Data Availability

The raw data supporting the conclusions of this article will be made available by the authors upon request.

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
