# Peer review of "In Vitro Mucoadhesive Features of Gliadin Nanoparticles Containing Thiamine Hydrochloride"

_pharmaceutics, 2024, doi:10.3390/pharmaceutics16101296_

Round 1
Reviewer 1 Report
Comments and Suggestions for Authors
This manuscript provides valuable insights into the use of gliadin nanoparticles as oral drug delivery systems, focusing on the surface properties of the carrier, in vitro adhesion ability to the gastric mucosal layer, and drug encapsulation and release characteristics. However, I personally recommend that the authors shall further explore the stability and biocompatibility of this nanoparticle under simulated gastrointestinal conditions (such as within the acidic medium with digestive enzymes.) Therefore, this additional studies shall be added to make it more scientifically rigorous.
Comments on the Quality of English LanguageThe quality of the English Language is fine to me.
Reviewer 2 Report
Comments and Suggestions for Authors
Review for the publication: In vitro mucoadhesive features of gliadin nanoparticles containing thiamine hydrochloride
Dear Authors,
Thank you for the submitted manuscript. It presents an interesting topic; however, it requires revisions and corrections to improve its scientific quality and overall reception. Below are my comments:
- The abstract should be expanded with 2 to 3 sentences of theoretical background at the beginning and supplemented with obtained data and results. Ideally, the results should include numerical values—this will enhance the article's citation potential, as readers will be able to quickly assess whether the presented data are of interest to them.
- The introduction lacks a clear emphasis on what the novelty of the presented work is. While the authors have provided an interesting introduction to the topic, there is a need to highlight the specific research niche more clearly. Please address this.
- I suggest adding a diagram of the preparation process to Section 2.2.
- Please specify which ANOVA was used in the statistical analysis.
- Please prepare a table describing the compositions of the various materials that were analyzed.
- Please prepare a table describing the characteristic groups appearing in the FT-IR spectra.
I believe that the article is very well-prepared, and the authors have put a significant amount of effort into addressing the scientific issues described. Therefore, I think that after incorporating the information provided in the comments, the article could be published.
Reviewer 3 Report
Comments and Suggestions for Authors
In this article, authors have developed gliadin nanoparticles encapsulated with B1. Authors need to address following deficiencies or comments for suitability of research work for Pharmaceutics.
· Abstract need improvement.
· In introduction, authors claims in vitro interaction mucin and colloidal formulations is time saving and cost effective but for clinical application how formulations can be approved without animal testing?
· Is there any specific reason for using B1? What about fat-soluble vitamins? How B1 encapsulation in gliadin NPs will benefit consumers, when many over the counter formulations are available.
· Previous work on gliadin nanoparticles or B1 delivery via nanocarriers is missing in introduction section.
· In section 2.2, mention amount and phase in which stabilizer was used. Moreover, add a formulation chart in this for more clarity.
· No proper justification is given for selection of 0.6mg/ml formulation whereas formulation containing 0.2mg/ml is giving better entrapment and higher release of entrapped content.
· Why release was only limited to 6hrs? It does not simulates GIT conditions. Perform release upto 24 or 48 hrs so one can see if formulations could release the remaining entrapped content.
· At the end of section 3.1, clearly discuss and mention, which formulation is better and what are the basis? Therefore, readers have clearly understanding of formulation promoted for further studies in section 3.2.
· In figure 3a and 3 b keep the unit same to avoid the confusion. Better to use code for each formulation.
· Did authors developed 0.5mg/ml gliadin nanoparticles? It suddenly appeared in section 3.2, figure 5 and supplementary figure s3. Correct it.
· Provide stability data og developed formulations.
Comments on the Quality of English LanguageModerate editting
